# The Combined Effect of Pressure and Temperature on Kefir Production—A Case Study of Food Fermentation in Unconventional Conditions

**DOI:** 10.3390/foods9081133

**Published:** 2020-08-18

**Authors:** Ana C. Ribeiro, Álvaro T. Lemos, Rita P. Lopes, Maria J. Mota, Rita S. Inácio, Ana M. P. Gomes, Sérgio Sousa, Ivonne Delgadillo, Jorge A. Saraiva

**Affiliations:** 1LAQV-REQUIMTE, Department of Chemistry, University of Aveiro, 3810-193 Aveiro, Portugal; anacatarinaribeiro@ua.pt (A.C.R.); alvarotomaz@ua.pt (Á.T.L.); ritaplopes@ua.pt (R.P.L.); mjmota@ua.pt (M.J.M.); ritainacio@ua.pt (R.S.I.); ivonne@ua.pt (I.D.); 2CBQF—Centro de Biotecnologia e Química Fina, Laboratório Associado, Escola Superior de Biotecnologia, Universidade Católica Portuguesa, 4200-374 Porto, Portugal; amgomes@porto.ucp.pt (A.M.P.G.); sdcsousa2@gmail.com (S.S.)

**Keywords:** kefir, high pressure, temperature, fermentation kinetics

## Abstract

Food fermentation under pressure has been studied in recent years as a way to produce foods with novel properties. The purpose of this work was to study kefir production under pressure (7–50 MPa) at different temperatures (17–32 °C), as a case study of unconventional food fermentation. The fermentation time to produce kefir was similar at all temperatures (17, 25, and 32 °C) up to 15 MPa, compared to atmospheric pressure. At 50 MPa, the fermentation rate was slower, but the difference was reduced as temperature increased. During fermentation, lactic and acetic acid concentration increased while citric acid decreased. The positive activation volumes (Va) obtained indicate that pressure decreased the fermentation rate, while the temperature rise led to the attenuation of the pressure effect (lower Va). On the other hand, higher activation energies (Ea) were observed with pressure increase, indicating that fermentation became more sensitive to temperature. The condition that resulted in a faster fermentation, higher titratable acidity, and higher concentration of lactic acid was 15 MPa/32 °C. As the authors are aware, this is the second work in the literature to study the combined effect of pressure and temperature on a fermentative process.

## 1. Introduction

Kefir is a lightly carbonated, low-alcohol dairy beverage with a creamy consistency and a pH ≈ 4.2–4.6 [1,2,3] that is produced by milk fermentation with starter cultures or kefir grains that contain lactic acid bacteria; yeasts; and, in some cases, acetic acid bacteria [4,5]. Kefir quality is affected by different fermentation parameters, such as type of kefir culture, inoculation ratio, time, and temperature [1]. The fermentation temperature distinctly influences the final acidity value, acid production rate, and the fermentative rate [6]. For instance, Dimitreli and Antoniou (2011) [7] found that a lower fermentation temperature led to a time extension needed to produce kefir (pH 4.4) due to the slower microbial activity at lower temperatures [8].

High pressure (HP) is a non-thermal technology mostly used in food preservation as a non-thermal pasteurization process [9]. Recently, new applications for HP have been studied, such as the use of sub-lethal pressures (5–100 MPa) in the microbial fermentation processes [10,11,12,13,14,15,16]. Although only few studies have been published about fermentation under HP, it was reported that when microorganisms are under these levels of pressure they can develop specific stress response mechanisms, such as metabolic modulation [12] and/or modulation of gene expression [17,18]. Picard et al. (2007) [11] performed alcoholic fermentation by *Saccharomyces cerevisiae* under HP (5–100 MPa at 30 °C), noting an increase in fermentation rate with the pressure increase up to 10 MPa, reaching a maximum ethanol production at 5 MPa. Likewise, glycerol fermentation by *Lactobacillus reuteri* under HP (10–35 MPa at 37 °C) exhibited an increase in the production of 1,3-propanediol at 10 MPa [12]. Another study investigating probiotic yogurt production under HP (5–100 MPa at 43 °C) reported lower fermentation rates with increased pressure, but the extension of the fermentation time at 5 MPa revealed that it was still possible to produce yogurt [13]. Moreover, the combination of pressure and temperature (10 and 30 MPa at 25–50 °C) resulted in a higher fermentation rate at 10 and 30 MPa/43 °C, with the fastest fermentation conditions observed at 10 MPa/43 °C [14]. A more recent work by Vieira (2019) [10] found that pressure increased the firmness of yoghurt and decreased plastic adhesiveness, whereas an overall brief sensorial analysis revealed that the yogurts produced at 10 and 20 MPa were preferred over the yogurt produced at atmospheric pressure.

In literature, the application of HP related to kefir was only used either before or after fermentation, in the former case to study the effect on microbiota for subsequent fermentation [19] and in the latter situation to preserve kefir [20]. Another possibility is to apply HP throughout the fermentation process to study its effect on the fermentation process, namely, at a first instance on kinetics of fermentation. Thus, the present work studied the combined effect of pressure (7, 15, 30, and 50 MPa) and temperature (17, 25, and 32 °C) on kefir kinetics, being one of the very few works in the literature that is related to the combined effect of pressure and temperature in a fermentation process and that uses kefir as a case study (the other case studied thus far used yogurt).

## 2. Materials and Methods

### 2.1. Sample Preparation

Commercial whole milk powder (Nestlé, Portugal) was reconstituted at 10% (*w*/*v*) and pasteurized (90 °C, 20 min) [8], followed by inoculation with commercial lyophilized culture kefir (A-Vogel, Switzerland) in a proportion of 0.7 g for 1 L of milk, as indicated on the label. The samples were packed under aseptic conditions in polyamide-polyethylene bags (Plásticos Macar Lda., Santo Tirso, Portugal), previously sterilized with UV radiation, which were manually heat sealed to minimize the amount of air inside the bags. The inoculum was composed of both bacteria (*Lactococcus lactis* subsp. *lactis*, *Lactococcus lactis* subsp. *cremoris*, *Lactococcus lactis* subsp. *lactis biovar diacetylactis*, *Leuconostoc mesenteroides* subsp. *cremoris*, *Lactobacillus acidophilus*, *Streptococcus thermophilus*, *Lactobacillus kefyr*) and yeasts (*Kluyveromyces marxianus* var. *marxianus* and *Saccharomyces unisporus*).

### 2.2. Kefir Fermentation under HP

Initially, fermentation was performed at 17 °C (used as control temperature in this work) for 28 h and at different pressure conditions (7, 15, 30, and 50 MPa). These experiments were conducted in an HP equipment (SFP FPG13900, Stansted Fluid Power Ltd., Harlow, UK), using a mixture of propylene glycol and water (40:60) as pressurizing fluid. Higher temperature experiments (25 and 32 °C, for 72 and 32 h, respectively) were carried out at 15 and 50 MPa, using a different HP equipment (High pressure system U33, Unipress Equipment, Poland) and the same mixture as pressurizing fluid. As a control, fermentation was carried out at 0.1 MPa (atmospheric pressure) at each temperature. The samples were immersed in the same fluid and in the dark to create similar conditions to the experiments under pressure. For all fermentation conditions, duplicated samples were studied and collected throughout the fermentation time and the analyses were also performed in duplicate.

### 2.3. pH Values and Titratable Acidity

The pH was measured at 22 °C with a properly calibrated pH glass electrode (Crison Instruments, S. A., Barcelona, Spain) by directly submerging the probe into the homogenized kefir samples.

Titratable acidity was quantified according to Chandan et al. (2006) [21], with some modifications, by titrating of 12 mL of diluted kefir to pH = 8.9 with a 0.1 N NaOH solution (Merck KGaA, Darmstadt, Germany) using an automatic titrator (Titromatic 1S, Crison Instruments, S.A., Barcelona, Spain). The results were expressed in milligrams of lactic acid/g of kefir.

### 2.4. Reducing Sugars Concentration

The reducing sugars concentrations were determined by applying the colorimetric method of 3,5-dinitrosalicylic acid reagent (DNS) (Acros Organics, Geel, Belgium) [22]. Briefly, 1 mL of DNS reagent was added to 1 mL of sample and slightly shaken. The samples were then placed in a boiling water bath for 5 min, cooled on ice, and diluted with 10 mL of distilled water. The absorbance of samples was measured at 540 nm using a Multiskan GO Microplate Spectrophotometer (Thermo Fisher Scientific Inc., NJ, USA). The concentration values were calculated by a calibration curve using glucose as standard (0–1.0 g/L, y = 0.524x − 0.027, *R*^2^ = 0.997) and the results are expressed in milligrams of reducing sugars/g of kefir.

### 2.5. Activation Volume and Activation Energy Calculation

The fermentation rate constants (*k*) for H^+^ concentration, titratable acidity, and reducing sugar concentration were estimated using the window where linear variation of the curves occurred (in the initial part of the fermentation process). For example, *k* was obtained by plotting the ln (H^+^ concentration) versus fermentation time, where *k* is the slope of the linear form. The *k* values were then used to estimate the activation volume (Va) and activation energy (Ea) using linear forms of the Eyring Law (Equation (1)) and the Arrhenius Law (Equation (2)):(1)$$\ln\left(k\right)=\ln\left(A\right)-\;\mathrm{Va}\times \frac{p}{R\times T}$$
(2)$$\ln\left(k\right)=\ln\left(A\right)-\;\mathrm{Ea}\times \frac{1}{R\times T}$$
where *k* is the fermentation rate constant (h^−1^), A is a constant, Va is the activation volume (cm^3^/mol), Ea is the activation energy (kJ/mol), p is the pressure (MPa), R is the universal gas constant (8.314 cm^3^·MPa/K·mol or 8.314 J/K·mol), and T is the absolute temperature (K).

### 2.6. Organic Acids, Sugars, and Ethanol Determination

Extraction of organic acids and sugars of kefir samples were performed following the method described by Costa et al. (2016) [23] with some modifications. Briefly, 1 g of homogenized kefir samples was added to 5 mL of H_2_SO_4_ (45 mmol/L) (Fisher Scientific, UK) for 1 min in a vortex and the mixture was then stirred in a multi-purpose rotator PSU-10i (Biosan, Latvia) for 30 min at 240 rpm. The homogenates were centrifuged at 6000 rpm for 30 min at 4 °C (Centurion Scientific, Scansci, LDA, Vila Nova de Gaia, Portugal) and the supernatant was filtered through a 0.22 μm pore size membrane filter and stored at −20 °C until HPLC analysis.

The chromatographic system consisted in a HPLC Knauer system equipped with Knauer K-2301 RI detector and an Aminex HPX 87H cation exchange column (300 × 7.8 mm) (Bio Rad Laboratories Pty Ltd., CA, USA). The mobile phase was 13 mM H_2_SO_4_, delivered at a flow rate of 0.6 mL·min^−1^ and the column kept at 65 °C. Peaks were identified by their retention times and quantified using calibration curves prepared with different standards.

## 3. Results and Discussion

### 3.1. Effect of Pressure Fermentation on Kefir Production

#### 3.1.1. Fermentation at Control Temperature

For fermentation at control temperature (17.0 °C), the results of pH variation, titratable acidity, and reducing sugars concentration are shown in Figure 1. Regarding pH values, a gradual decrease over time was seen, reaching a final pH value of 4.35 ± 0.04 after 24 h of fermentation at 0.1 MPa, which corresponds to the typical pH range of kefir (4.2–4.6) [3]. With pressure increase there was a slower decrease of pH, i.e., a lower fermentation rate, according to the results obtained when producing probiotic yoghurt under pressure [13]. Furthermore, at 7 and 15 MPa, it was possible to produce kefir with pH values of 4.53 ± 0.02 and 4.57 ± 0.03, respectively, after the same fermentation time as for fermentation at 0.1 MPa (24 h). These results agree with those of Bothun et al. (2004) [16] who reported that *Clostridium thermocellum* can perform fermentation at 7 MPa, producing ethanol from cellulosic material. Differently, Mota et al. (2015) [13] observed that an extension of the fermentation time under pressure (5 MPa) was necessary to achieve probiotic yoghurt production. For fermentation at higher pressures (30 and 50 MPa), an even slower decrease on pH was verified, similar to what was reported by Mota et al. (2015) [13] and Picard et al. (2007) [11], and the typical pH of kefir was not achieved after 28 h.

The results of titratable acidity agree with those obtained for pH, except for fermentation at 15 MPa, since fermentations performed at 7 and 15 MPa presented a similar pH value after 24 h, but the titratable acidity values were lower for 15 MPa (Figure 1a). The reducing sugars concentration (Figure 1b) showed a slower decrement rate and a higher remaining value after 28 h, with increasing pressure. These might result from the progressive suppression of enzymatic activity involved in the glycolytic pathway as pressure increases. For example, inhibition of the phosphofructokinase is reported in the literature due to induction of acidification by pressure at about 50 MPa [24,25].

#### 3.1.2. Fermentations at Temperatures above Control Temperature

After the study of fermentation at control temperature, we studied the effect of pressure at different fermentation temperatures. For this purpose, two temperatures higher than 17 °C were selected (25 and 32 °C), avoiding temperatures below 17 °C that could result in a much slower fermentation process, thus delaying the time needed for the experiments. Regarding pressure, two levels were selected: a lower level of pressure, where fermentation and kefir production was seen (15 MPa), and a higher level of pressure, where the fermentation was almost inexistent at 50 MPa at the control temperature.

For 25 °C, the results for pH variation and titratable acidity are shown in Figure 2, where it can be seen that fermentation at 0.1 and 15 MPa exhibited a similar profile of pH variation over time. The results showed that kefir production occurred at all pressures tested: (i) after 8 h for fermentations at 0.1 and 15 MPa (pH = 4.55 ± 0.03 and 4.57 ± 0.02, respectively) and (ii) after 32 h of fermentation at 50 MPa (pH = 4.43 ± 0.01). Titratable acidity results are in accordance with pH variation, reaching a similar titratable acidity after 72 h of fermentation at 0.1 and 15 MPa, while a lower level of titratable acidity was seen for 50 MPa. However, when the kefir pH was reached for all pressures, the titratable acidity was similar (about 7 mg/g).

For fermentation at 32 °C, the results of pH variation and titratable acidity are shown in Figure 3. Fermentations at 0.1, 15, and 50 MPa revealed a similar profile in pH variation, with 50 MPa shown to affect the fermentation to a lesser extent compared to 25 °C. In all cases, kefir was obtained but the time required for kefir production increased with pressure increment: (i) at 0.1 and 15 MPa after 6 h of fermentation (pH = 4.56 ± 0.05 and 4.66 ± 0.02, respectively) and (ii) at 50 MPa after 24 h (pH = 4.17 ± 0.01). Regarding titratable acidity, similar results were obtained for 0.1 and 15 MPa over the fermentation time and a lower value was achieved for fermentation at 50 MPa. A possible explanation for the results at 50 MPa could be hypothesized as the production of a relatively higher amount of acid, whose pKa values may have been higher than final pH of kefir, thus being less ionized and therefore contributing less to pH change while being accountable for titratable acidity, possibly due to metabolic changes caused by pressure.

These results point out that while fermentation occurred at the same rate of atmospheric pressure at a lower pressure level (15 MPa), at both temperatures (25 and 32 °C), when pressure was increased to higher values (50 MPa), the pH decrease rate was higher at 32 °C. This shows that the effect of fermentation slowing down caused by increasing pressure was counteracted by temperature, with pressure and temperature showing an antagonistic effect. It is still important to note that it was possible to produce kefir even at the higher pressure studied (50 MPa), contrarily to what occurred at the control temperature at the same pressure.

#### 3.1.3. Kinetic Analysis

##### Fermentation Rate Constants

The fermentation rate constants for H^+^ concentration, titratable acidity, and reducing sugars concentration are presented in Table 1. Overall, at 17 °C, the *k* values corresponding to H^+^ concentration/titratable acidity decreased when pressure increased; in particular, for 30 and 50 MPa, the rates decreased (compared to 0.1 MPa) by 1.97-/2.17- and 6.48-/8.33-fold, respectively, showing a higher effect of pressure on titratable acidity.

For 25 and 32 °C, the *k* values for H^+^ concentration revealed that pressure increment led to a lower reduction in fermentation rate compared to 17 °C. For H^+^ concentration at 0.1 MPa, fermentation rates were 1.12- and 1.11-fold higher than at 15 MPa (for 25 and 32 °C, respectively). For titratable acidity, fermentation at 15 MPa (0.116 and 0.159 h^−1^ for 25 and 32 °C) showed a slightly higher/equal *k* value compared to 0.1 MPa (0.099 and 0.158 h^−1^ for 25 and 32 °C), with the fermentation rate at 0.1 MPa being 0.85- and 0.99-fold lower than at 15 MPa. At 50 MPa, a higher decrease in titratable acidity rate was observed for the three temperatures, but was more pronounced for 17 °C than for 25 and 32 °C, as for the former, at 0.1 MPa, the rate was 8.33-fold higher, while for the latter two was higher only by 2.48- and 2.16-fold, respectively.

##### Activation Volume

The activation volume (Va) is a kinetic parameter that gives information about the effect of pressure on reactions rates. For instance, a positive Va indicates a deceleration of the process by pressure increment, while a negative Va reveals its acceleration, which means that the higher the numerical value of Va, the greater the effect of pressure. In this study, Va values were calculated on the basis of the results of fermentation rate constants for H^+^ concentration, titratable acidity, and reducing sugars concentration (Table 2). For all cases, the Va values were positive, confirming that pressure slowed down the reactions involved in kefir fermentation, which is in a manner consistent with the results obtained for probiotic yogurt [13].

For fermentation at 17 °C, the reducing sugars concentration was the parameter with the lowest Va value (77.72 cm^3^/mol), indicating that reactions involved in sugar consumption were less affected by pressure, which can be related to the possibility of sugars being used not only for the fermentative process but also for adaptation to pressure. These results are in accordance with the results of Iwahashi et al. (2005) [17] and Bravim et al. (2013) [18], who observed that *S. cerevisiae* subjected to pressure treatments exhibited an increase of gene expression involved in metabolism of carbohydrates and stress response, and was related to hexose transporters and encoding glycolytic enzymes. On the other hand, the Va value estimated for titratable acidity was 96.88 cm^3^/mol, with this parameter being more sensitive and thus more affected by pressure. These results agree with what was discussed above in relation to the possibility of the production of higher amounts of acids under pressure, which contributed less to pH change while accounting for increased titratable acidity, possibly due to metabolic changes caused by pressure.

For fermentation at 25 °C, similar Va values were achieved for H^+^ concentration and titratable acidity (52.46 and 50.45 cm^3^/mol, respectively), while for 32 °C, titratable acidity was slightly more affected by pressure than H^+^ concentration (42.33 and 37.76 cm^3^/mol, respectively).

Hence, overall, the titratable acidity was the parameter more affected by pressure and the temperature increase induced a reduction in Va values, evidencing an attenuation of the pressure effect on slowing down the fermentation (interestingly, quantitatively Va showed a ln-linear trend decrease with temperature (y = −0.053 + 5.32, *R^2^* of 0.996 and y = −0.056 + 5.46, *R^2^* of 0.922, respectively, for H^+^ concentration and titratable acidity). This attenuation may be related to the production of heat shock proteins that occurs as a stress response in many microorganisms. For instance, Aertsen et al. (2004) [26] reported that in *Escherichia coli* the induction of several heat shock genes occurred after exposure to sub-lethal pressures. These authors concluded that heat shock proteins may play a key role in preventing cellular damage and/or helping the cell recovery, which may have occurred when the kefir fermentation was carried out in the different pressure/temperature combinations.

### 3.2. Effect of Fermentation Temperature on Kefir Production

#### 3.2.1. Fermentation at 0.1, 15, and 50 MPa

At 0.1 MPa, the increase of fermentation temperature from 17 °C (Figure 1a) to 32 °C (Figure 3) caused an increase on the fermentation rate, with the fermentation at 32 °C being about 3.67- and 1.34-fold faster than at 17 and 25 °C, respectively (Table 1). These results agree with the ones obtained by Apar et al. (2017) [27] and Dimitreli and Antoniou (2011) [7], who reported the same positive effect of temperature on kefir fermentation rate. The same behavior was observed under pressure (15 and 50 MPa), where an increase in the fermentation rate was observed with the increase of fermentation temperature (at 15 MPa/32 °C, the rate was 3.47- and 1.35-fold faster than at 17 and 25 °C, respectively). On the other hand, kefir production did not occur after 28 h at 50 MPa/17 °C, however, when the temperature was raised to 25 °C, kefir production took place after 32 h at 50 MPa. Additionally, at 32 °C, the reduction in fermentation time to produce kefir was more pronounced (24 h), since at this temperature the fermentation rate was 1.79-fold faster compared to 25 °C. These results indicate an antagonistic effect between pressure and temperature.

Regarding titratable acidity at 17 °C (Figure 1a), 25 °C (Figure 2), and 32 °C (Figure 3) for all pressures tested, it was possible to conclude that the temperature increase led to a higher final titratable acidity value, results that are in agreement with those of Irigoyen et al. (2003) [6]. Furthermore, at 50 MPa/32 °C, an increase of about 12.17-fold in the titratable acidity rate compared to 17 °C was verified, resulting in an increment in the final titratable acidity, reaching the maximum value at 32 °C for all pressures tested. These results are in line with those of Ismaiel et al. (2011) [28] for kefir production, where the authors showed that the maximum titratable acidity occurred at 35 °C in a temperature range between 15–50 °C (at 0.1 MPa).

#### 3.2.2. Kinetic Analysis

##### Activation Energy

The activation energy (Ea) is a kinetic parameter that gives information about the effect of temperature on the reaction rates. Thus, a higher/lower Ea value reveals that reactions were more/less temperature-sensitive, respectively, being calculated on the basis of the results of fermentation rate constants for H^+^ concentration and titratable acidity (Table 3).

At 0.1 MPa, the Ea values were 56.83 and 56.63 kJ/mol for H^+^ concentration and for titratable acidity, respectively, which shows that H^+^ concentration was slightly more sensitive to temperature. Conversely, for fermentation under pressure (15 and 50 MPa), titratable acidity was the parameter more sensitive to temperature (88.48/124.2 kJ/mol for 15 MPa/50 MPa, compared to 61.90/114.1 for H^+^ concentration). Therefore, the results suggest that an increase of pressure caused an overall increased sensitivity to temperature of the metabolic reactions involved in kefir fermentation. Similarly to Va, Ea showed a ln-linear trend increase with pressure (*y* = 0.015 + 3.98, *R^2^* of 0.967 and *y* = 0.015 + 4.13, *R^2^* of 0.905, respectively) for H^+^ concentration and titratable acidity.

### 3.3. HPLC Analysis of Sugars, Organic Acids, and Ethanol During Kefir Fermentation

Table 4 shows that sugar concentration (quantified at 17 °C) and a decrease in lactose concentration during the fermentation time was observed, but to a higher extent at 0.1 MPa (about 17% of consumption). When fermentation was performed under pressure, lactose concentration decreased by 13% at 15 MPa and about 10% at 7 and 50 MPa. Glucose concentration for fermentation at 0.1, 7, 15, and 50 MPa remained constant throughout the fermentation time, while galactose concentration increased during fermentation time for all conditions (except at 0.1 MPa/17 °C), with different variation profiles. Sugar concentration by HPLC followed the same trend as the reducing sugars, where there was also decrease in sugar over fermentation, but slightly greater at 0.1 MPa.

Lactic acid (Figure 4(a1,b1)) is the main acid produced in kefir fermentation by lactic acid bacteria and, as expected, while the pH decreased, this acid increased progressively with the fermentation and was the predominant acid present. Comparing 0.1 and 7 MPa, both at 17 °C, lactic acid concentration showed a slight tendency to be greater at 7 MPa (5.033 ± 0.109 mg/g versus 4.688 ± 0.488 mg/g). At 15 MPa/17 °C, the lactic acid reached a maximum value of 3.806 mg/g. However, at 50 MPa, production of this acid was very low as kefir production did not occur. At 32 °C and at 0.1 MPa, lactic acid reached a maximum value of 8.256 ± 0.326 mg/g, but pressure increment led to a decrease in lactic acid concentration at the end of fermentation to about 5.052 ± 0.136 mg/g.

Acetic acid (Figure 4(a2,b2)) at 17 °C was only measurable at 7 MPa after 24 h, while at 32 °C it could be quantified at all pressures tested, reaching a maximum value of 0.684 ± 0.071 mg/g (0.1 MPa). The presence of this acid in kefir could be attributed to heterofermentative lactic acid and acetic acid bacteria [29], since the cultures used in this study included no acetic acid bacteria.

Relative to citric acid (Figure 4(a3,b3)), the concentration presented in milk was 1.905 ± 0.163 and 2.643 ± 0.424 mg/g at 17 and 32 °C, respectively. For both temperatures, at the end of fermentation, the concentration was below the quantification limit at 0.1, 7, and 15 MPa, while at 50 MPa, acid concentration decreased by 4 and 51% at 17 and 32 °C, respectively.

Propionic, succinic, and formic acids were not detected in measurable amounts, which differs from other studies and may be the result of variations in the ratio and types of microorganisms in starter cultures of kefir [30].

Ethanol was also not detected, whereas in the literature, concentrations of 0.026–1.0% ethanol in kefir have been reported; however, this compound was not detected in kefir produced with starter cultures [31,32].

## 4. Conclusions

Kefir production under unconventional conditions of higher pressure showed that the time needed to produce kefir was similar at all temperatures (17, 25, and 32 °C) at 7 and 15 MPa compared to 0.1 MPa. However, for 50 MPa, the fermentation rate was slower and kefir production occurred only at 25 and 32 °C, while it was even faster at 15 MPa and 32 °C (also with higher final titratable acidity and higher concentration of lactic acid). In addition, the combination of pressure and temperature showed an antagonistic effect, and the inhibitory effect of pressure in fermentation was attenuated by a positive effect of temperature increment. The differences observed for pH variation, titratable acidity, and reducing sugars concentration under pressure indicated that microorganisms may trigger mechanisms of adaptation to pressure. Organic acid composition showed some differences in kefir produced under pressure, namely, the presence of a higher proportion of acids with pKa values higher than final pH of kefir at 50 MPa. Further studies are of interest to understand the combined effect of pressure and temperature on several characteristics of kefir, such as rheological, sensorial, nutritional, microbiological, and functional properties.

As the authors are aware, this is one of the first studies in the literature that reports the combined effect of pressure and temperature on a fermentative process, as well as the effect of these two variables in the activation energy and activation volume.

## Figures and Tables

**Figure 1 foods-09-01133-f001:**
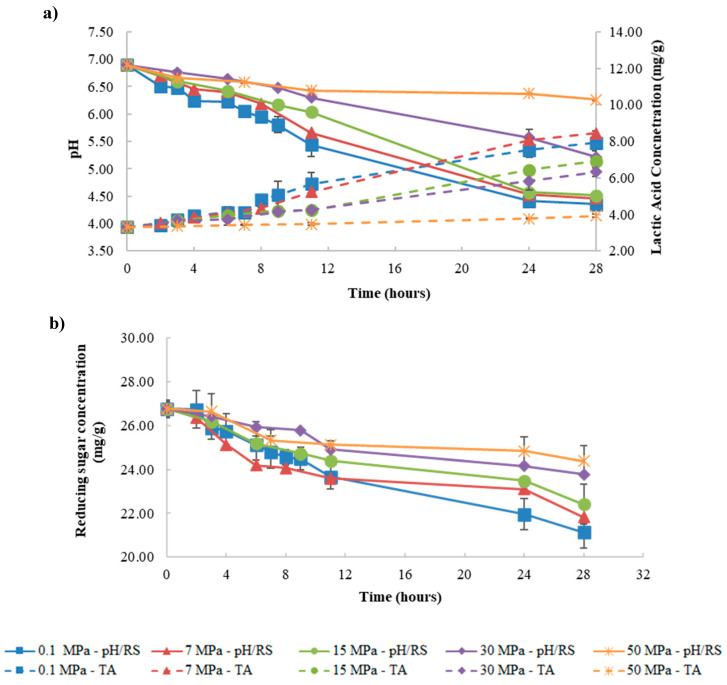
Variation of pH (solid lines) and titratable acidity (TA (broken lines), expressed as lactic acid concentration, mg/g) (**a**) and reducing sugar concentration (RS, mg/g) (**b**) during fermentation under different pressure conditions (0.1–50 MPa) at control temperature (≈17 °C).

**Figure 2 foods-09-01133-f002:**
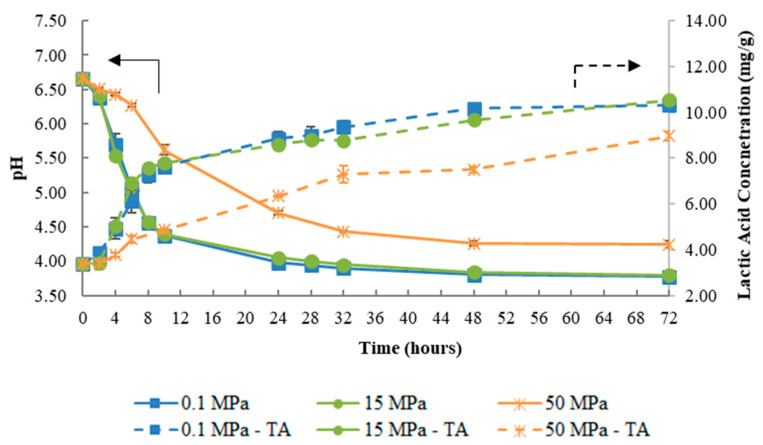
Variation of pH (solid lines) and titratable acidity (TA (broken lines), expressed as lactic acid concentration, mg/g) during fermentation under 0.1, 15, and 50 MPa at 25 °C.

**Figure 3 foods-09-01133-f003:**
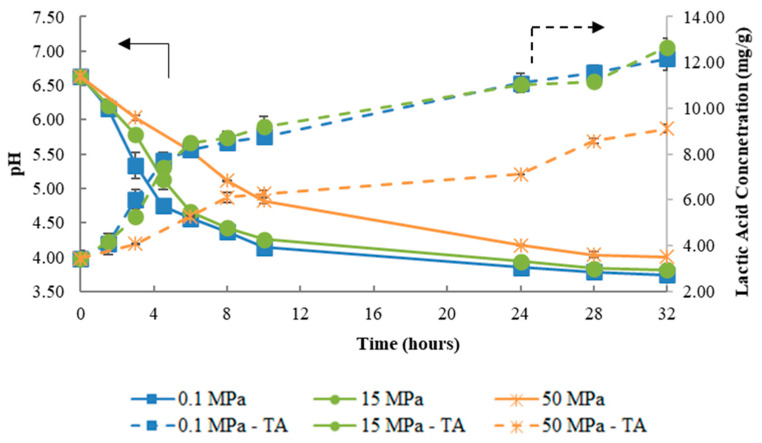
Variation of pH (solid lines) and titratable acidity (TA (broken lines), expressed as lactic acid concentration, mg/g), during fermentation under 0.1, 15, and 50 MPa at 32 °C.

**Figure 4 foods-09-01133-f004:**
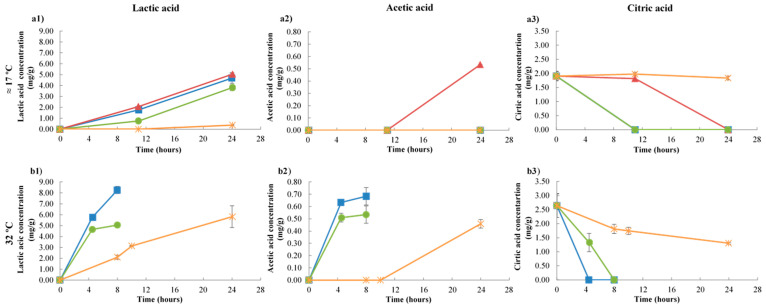
Variation of lactic (**a1**,**b1**), acetic (**a2**,**b2**), and citric (**a3**,**b3**) acid in kefir samples during fermentation under 0.1, 7, 15, and 50 MPa at ≈17 °C (**a**) and 32 °C (**b**).

**Table 1 foods-09-01133-t001:** Fermentation rate constants (k, h^−1^) for H^+^ concentration, titratable acidity, and reducing sugar concentration under different combinations of pressure (0.1–50 MPa) and temperature (17, 25, and 32 °C) during kefir production.

Temperature (°C)	Pressure (MPa)	Fermentation Rate Constant (*k*, h^−1^)		
		H^+^ Concentration	Titratable Acidity	Reducing Sugar Concentration
17	0.1	0.272 ^P^(−)/(3.14) ^T^	0.050 (−)/(3.16)	0.011
	7	0.230 (1.18)/(−)	0.040 (1.25)/(−)	0.015 (0.73)
	15	0.222 (1.23)/(3.47)	0.027 (1.85)/(5.89)	0.009 (1.22)
	30	0.138 (1.97)/(−)	0.023 (2.17)/(−)	0.004 (2.75)
	50	0.042 (6.48)/(9.95)	0.006 (8.33)/(12.17)	0.003 (3.67)
25	0.1	0.638 (−)/(1.34)	0.099 (−)/(1.60)	NE ^§^
	15	0.572 (1.12)/(1.35)	0.116 (0.85)/(1.37)	
	50	0.233 (2.74)/(1.79)	0.040 (2.48)/(1.83)	
32	0.1	0.854	0.158	NE
	15	0.771 (1.11)/(−)	0.159 (0.99)/(−)	
	50	0.418 (2.04)/(−)	0.073 (2.16)/(−)	

^P^ Ratio of the fermentation rate constants between the fermentation at 0.1 MPa and the other pressures tested (7, 15, 30, and 50 MPa), (k_0.1_/k_p_). ^T^ Ratio of the fermentation rate constants between the fermentation at 32 °C and the other temperatures tested (17 and 25 °C), (k_32°C_/k_T_). ^§^ NE means not evaluated.

**Table 2 foods-09-01133-t002:** Activation volumes (cm^3^/mol) of each physicochemical parameter analyzed for fermentation at 17, 25, and 32 °C.

Physicochemical Parameter	Temperature (°C)	Activation Volume (cm^3^/mol)–*R^2^*
**H^+^ concentration**	17	83.86–0.89
	25	52.46–0.96
	32	37.76–0.97
**Titratable acidity**	17	96.88–0.94
	25	50.45–0.82
	32	42.33–0.91
**Reducing sugar concentration**	17	77.72–0.88

**Table 3 foods-09-01133-t003:** Activation energies (kJ/mol) of each physicochemical parameter analyzed for fermentation at 0.1, 15, and 50 MPa.

Physicochemical Parameter	Pressure (MPa)	Activation Energy (kJ/mol)–*R^2^*
**H^+^ Concentration**	0.1	56.83–0.95
	15	61.90–0.94
	50	114.1–0.95
**Titratable Acidity**	0.1	56.63–0.99
	15	88.48–0.91
	50	124.2–0.94

**Table 4 foods-09-01133-t004:** Lactose, glucose, and galactose concentration quantified by HPLC analysis during fermentation under different pressure conditions (0.1, 15, and 50 MPa) at 17 °C (the analysis was carried out at the beginning and the end of the fermentation).

Conditions	Lactose (mg/g)		Substrate Consumption (%)	Glucose (mg/g)		Galactose (mg/g)	
	0 h	24 h		0 h	24 h	0 h	24 h
**0.1 MPa**	32.49 ± 1.47	27.09 ± 2.14	17	0.99 ± 0.01	0.99 ± 0.04	BQL ^‡^	BQL
**7 MPa**		29.37 ± 1.65	10		1.03 ± 0.10		0.31 ± 0.05
**15 MPa**		28.37 ± 2.64	13		0.94 ± 0.12		0.22 ± 0.02
**50 MPa**		29.32 ± 0.58	10		1.16 ± 0.16		0.41 ± 0.10

^‡^ BQL means below the quantification limit.

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
