# Peer review of "The Combined Effect of Pressure and Temperature on Kefir Production—A Case Study of Food Fermentation in Unconventional Conditions"

_foods, 2020, doi:10.3390/foods9081133_

Round 1
Reviewer 1 Report
The current article has focused on the evaluation of the effect of pressure and temperature on kefir fermentation and the evaluation of different physicochemical characteristics with retrospective methods.
Although the different measurements taken could have given useful information about the effect of pressures on the physicochemical characteristics of Kefir fermentation, the experimental design of the study is insufficient.
Some of the major drawbacks of the study are: The experiment was performed only one time with one lot per case (one batch) and no replicate samples are reported per case. So the results cannot be reliable.
Again, as the experiment was not repeated 2 times and no replicate samples were used and the results are biased.
There are no microbiological studies to study the effect of pressure to the inoculum on the different cases. In addition, it would be important to study the different (if any) sensory characteristics and the acceptance of the final product.
Reviewer 2 Report
Foods (ISSN 2304-8158)
Manuscript ID. foods-876324
The combined effect of pressure and temperature on kefir production – A case-study of food fermentation in unconventional conditions
Ana C. Ribeiro, Álvaro T. Lemos , Rita P. Lopes , Maria J. Mota , Rita S. Inácio , Ana M. P. Gomes , Sérgio Sousa , Ivonne Delgadillo , Jorge A. Saraiva *
Abstract
Food fermentation under pressure is being studied in the last years as a possibility to produce foods with novel properties. Kefir production under pressure (7-50 MPa) at different temperatures (17-32 °C) was studied in this work, as a case-study of unconventional food fermentation. The fermentation time to produce kefir was similar at all temperatures (17, 25 and 32 °C) up to 15 MPa, compared to atmospheric pressure. At 50 MPa the fermentation rate was slower, but the difference was reduced as temperature increased. During fermentation, lactic and acetic acids concentration increased, while citric acid decreased. The positive activation volumes (Va) obtained indicate that pressure decreased the fermentation rate, while the temperature rise led to the attenuation of the pressure effect (lower Va). On the other hand, higher activation energies (Ea) were observed with pressure increase, indicating that the fermentative process became more sensitive to temperature. The combination that resulted in a faster fermentation, a higher titratable acidity, and a higher concentration of lactic acid was 15 MPa/32 °C. As the authors are aware, this is the second work in the literature to study the combined effect of pressure and temperature on a fermentative process.
It is a topic of interest to the researchers in the related area but the paper needs minor improvements. My detailed comments are as follows:
- The introduction, materials and methods in the paper work very well, especially the part that correspond to the Ph-VALUES AND TITRATABLE ACIDITYs.
- Results are good, and the figures are clear. The combination of results and discussion is very well done. it is very easy to follow the script of the work.
- In the conclusions section. I consider that it would be necessary to indicate more relevant aspects with the good experimental results. The new characteristics of kefir should be studied in comparison with the current ones. (It would be interesting then to make references to previous works in reference to organoleptic properties).
- The language is not fluent, suggesting that the paper should be language-edited by native english-speaker editor or colleagues. Please make minor revisions, especially in the part of dicussion.
Round 2
Reviewer 1 Report
Line 520 add also "microbiological studies"
Author Response
The amendment "Line 520 add also "microbiological studies" was changed accordingly.